# *Lacticaseibacillus rhamnosus* CA15 (DSM 33960) as a Candidate Probiotic Strain for Human Health

**DOI:** 10.3390/nu14224902

**Published:** 2022-11-19

**Authors:** Alessandra Pino, Amanda Vaccalluzzo, Cinzia Caggia, Silvia Balzaretti, Luca Vanella, Valeria Sorrenti, Aki Ronkainen, Reetta Satokari, Cinzia Lucia Randazzo

**Affiliations:** 1Department of Agricultural, Food and Environment, University of Catania, Santa Sofia Street, 100, 95123 Catania, Italy; 2ProBioEtna SRL, Spin Off of the University of Catania, Santa Sofia Street, 100, 95123 Catania, Italy; 3CERNUT, Interdepartmental Research Centre in Nutraceuticals and Health Products, University of Catania, A. Doria Street, 6, 95125 Catania, Italy; 4Sacco Systems, Alessandro Manzoni Street, 29/A, 22071 Cadorago, Italy; 5Department of Drug and Health Science, University of Catania, A. Doria Street, 6, 95125 Catania, Italy; 6Human Microbiome Research, Faculty of Medicine, University of Helsinki, Haartmaninkatu 3, FI-00014 Helsinki, Finland

**Keywords:** lactobacilli, functional features, safety, pathogen inhibition, antioxidant activity, anti-inflammatory activity

## Abstract

Lactobacilli with probiotic properties have emerged as promising tools for both the prevention and treatment of vaginal dysbiosis. The present study aimed to study the in vitro probiotic potential of the *Lacticaseibacillus rhamnosus* CA15 (DSM 33960) strain isolated from a healthy vaginal ecosystem. The strain was evaluated for both functional (antagonistic activity against pathogens; H_2_O_2_, organic acid, and lactic acid production; antioxidant and anti-inflammatory activities; ability to adhere to intestinal mucus and to both CaCo-2 and VK7/E6E7 cell lines; exopolysaccharide production; surface properties; and ability to survive during gastrointestinal transit) and safety (hemolytic, DNase, and gelatinase activities; mucin degradation ability; production of biogenic amines; and resistance to antimicrobials) characteristics. Data revealed that the tested strain was able to antagonize a broad spectrum of vaginal pathogens. In addition, the adhesion capacity to both vaginal and intestinal cell lines, as well as anti-inflammatory and antioxidant activities, was detected. The ability of the *Lacticaseibacillus rhamnosus* CA15 (DSM 33960) strain to survive under harsh environmental conditions occurring during the gastrointestinal passage suggests its possible oral delivery. Thus, in vitro data highlighted interesting probiotic properties of the CA15 (DSM 33960) strain, which could represent a valuable candidate for in vivo vaginal infections treatment.

## 1. Introduction

Vaginal microbiota, although characterized by few microbial species, is a dynamic ecosystem subjected to changes during a woman’s lifetime, from birth to pre- and postmenopause [1,2,3]. In addition, intrinsic (e.g., race, immune system imbalance, genetic susceptibility, etc.) and extrinsic factors (such as a diet rich in vitamins and folic acid, use of oral contraceptives, sexual behaviors, hygiene habits, antibiotics, or immunosuppressor therapies) can affect the balance of the vaginal bacterial biota. It is well known that the reduction in lactobacilli and the increase in facultative or obligate anaerobic microorganisms, along with different *Candida* species, are associated with vaginal microbiota dysbiosis [4,5]. Although most lactobacilli have a GRAS (generally recognized as safe) status and some of them are included in the Qualified Presumption of Safety (QPS) list [6,7], both safety and probiotic features are strain-specific and cannot be generalized to the species. Several studies demonstrated the ability of probiotic lactobacilli to protect vaginal health by different mechanisms of action, including antagonistic activity against pathogens through competitive exclusion, competition for nutrients, and production of metabolites with antimicrobial properties. In particular, the antagonistic activity against pathogens, one of the key features of probiotics, could be linked to the ability of lactobacilli to acidify the vaginal environment, producing mainly lactic acid thus preventing the colonization and proliferation of pathogens [4,5]. In addition, the ability to adhere to epithelial cells and to form biofilm on vaginal mucosa as well as modulation of the immune system is included among the health-promoting properties of lactobacilli [7,8,9,10,11,12,13,14,15]. In fact, the ability to adhere to the vaginal epithelium and compete for adhesion sites is involved in the inhibition of colonization by pathogens. In this context, the formation of a biofilm is a strategy that allows organisms to persist under stressful conditions allowing resistance to harsh environments. In addition, the development of a biofilm, by promoting mucosal colonization, interferes with the growth and adhesion of pathogens [16].

The beneficial effect of lactobacilli to balance vaginal microbiota has recently gained a lot of interest. This assumption is corroborated by several clinical studies reporting their usefulness for both the prevention and treatment of urogenital tract infections in women under different physiological conditions [2]. According to that, the use of lactobacilli is considered a valuable approach for the prevention and treatment of urogenital tract infections [2]. Probiotic strains, ascribed to the *Lacticaseibacillus rhamnosus* species, were extensively studied to evaluate their potential in the treatment of vaginal dysbiosis. Based on clinical applications, the *L. rhamnosus* GR-1 strain was firstly studied for the ability to balance the vaginal microbiota reducing the recurrence of urinary tract infections (UTIs) [9]. As reported by De Alberti and coworkers [17], *L. rhamnosus* HN001 and *L. acidophilus* La-14, orally administrated, significantly increased the vaginal *L. rhamnosus* and *L. acidophilus* abundance. Similarly, when administrated in association with *L. acidophilus* GLA-14, the *L. rhamnosus* HN001 strain significantly inhibited the cell density of *Gardnerella vaginalis*, *Atopobium vaginae*, *Staphylococcus aureus*, and *Escherichia coli* [18]. Recently, Pino and coworkers [19], demonstrated, by conducting an in vivo pilot study, the ability of the *L. rhamnosus* TOM 22.8 strain to restore normal vaginal microbiota in patients with bacterial vaginosis (BV).

Although commercial probiotic products are now available, there is great interest from nutraceutical companies in selecting new probiotic candidates to benefit the vaginal environment. In the present study, the probiotic potential, as well as the safety properties of the *Lacticaseibacillus rhamnosus* CA15 (DSM 33960) strain isolated from a healthy woman of reproductive age, was in vitro investigated.

## 2. Materials and Methods

### 2.1. Strain Isolation and Taxonomic Identification

The *Lacticaseibacillus rhamnosus* CA15 (DSM 33960) strain, belonging to the culture collection of ProBioEtna srl (spinoff of the University of Catania), was isolated from the vaginal ecosystem of a healthy Italian woman, of reproductive age, at the Department of General Surgery and Medical Surgical Specialties, General Hospital G. Rodolico, University of Catania (Catania, Italy) who was recruited to an observational study approved by the Local Ethical Committee (n. 16/2022/PO). Vaginal swab samples were collected, transported to the Laboratory of Microbiology, Department of Agricultural Food and Environment, University of Catania, Catania, Italy, and immediately processed as previously reported [15,19]. The *L. rhamnosus* CA15 (DSM 33960) strain was subjected to cell morphology evaluation and tested for both catalase activity and Gram reaction before genetic identification.

### 2.2. Species Identification and PFGE Analysis

A stationary culture of the *L. rhamnosus* CA15 (DSM 33960) strain was subjected to automated genomic DNA extraction using a dedicated kit according to the manufacturer’s instructions for Gram-positive bacteria (Qiagen, Germany). The genomic DNA preparation was subsequently used as template for a PCR using the primers P1–P4 [20] targeting the V1–V3 regions of the 16S rDNA gene [21,22,23]. The resulting amplicon was sequenced, and the species was identified against the BLAST “Nucleotide collection (nt/nt)” database (http://blast.ncbi.nlm.nih.gov/Blast.cgi, (accessed on 18 February 2022) gated to the “Type Strains”. The identification was considered univocal if the sequence homology was >99% [23,24]. The isolate typing was performed by Pulsed Field Gel Electrophoresis (PFGE), using the CHEF Bacterial Genomic DNA Plug Kits (Bio-Rad, Milan, Italy) according to the protocol indicated by the manufacturer for Gram-positive bacteria. Restriction enzyme digestion and electrophoresis were performed as described by Coudeyras and coworkers [25]. After staining with 1× Atlas ClearSight DNA Stain (Bioatlas, Tartu, Estonia) solution for one hour, the gels were visualized under UV light.

### 2.3. Safety Features

#### 2.3.1. Susceptibility to Antibiotics and Hemolytic, DNAse, Gelatinase, and Mucin Degradation Activities

The *L. rhamnosus* CA15 (DSM 33960) strain was tested for susceptibility to antibiotics following the European Food Safety Authority (EFSA) guidelines [26]. For each tested antibiotic, the minimal inhibitory concentration (MIC) value was determined by the broth microdilution method according to the ISO 10932:2010 procedure [27].

The hemolytic, DNAse, and gelatinase activities as well as the mucin degradation ability of the *L. rhamnosus* CA15 (DSM 33960) strain were tested following the protocol reported by Pino and coworkers [28]. *Streptococcus pyogenes* ATCC 19615 and *Streptococcus pneumoniae* ATCC 6303, previously cultured on Brain Heart Infusion broth (BHI, Becton Dickinson GmbH, Germany) at 37 °C under 5% of CO_2_, were used as positive controls for β-hemolysis and α-hemolysis, respectively. All analyses were performed in triplicate.

#### 2.3.2. Biogenic Amine Production

The *L. rhamnosus* CA15 (DSM 33960) strain was reactivated by inoculation at 2% into de Man Rogosa and Sharp medium (MRS, Oxoid, Milan, Italy) and incubated at 37 °C overnight. The resulting biomass was subcultured five times in decarboxylase medium broth supplemented with 0.005% of pyridoxal-5′-phosphate (Sigma-Aldrich, St. Louis, MO, USA), and the production of biogenic amines (BA) was stimulated by adding either 0.1% histidine, 0.1% lysine, 0.1% tyrosine, or 0.1% ornithine (Sigma-Aldrich, St. Louis, MO, USA) as described by Bover-Cid and Holzapfel [29]. Biogenic amine extraction and estimation protocols were performed as described previously [30]. Specifically, extracted and derivatized samples were separated by HPLC (Shimadzu Nexera XL UHPLC system equipped with a SPD M30A diode array detector) using a 250 mm × 46 mm × 5µm Discovery HS C18 column (Sigma-Aldrich) applying a low-pressure gradient of acetonitrile and water. Unlike the reference paper, the separation temperature was kept constant at 26 °C to allow a better reproducibility of the results. Six-point standard curves were generated and considered suitable if they presented a correlation R2 value of >0.99. The unknown samples were assayed in duplicate, and the results were analyzed as average ± standard deviation.

### 2.4. In Vitro Functional Features

#### 2.4.1. Antagonistic Activity against Pathogens

The pathogenic strains *Enterobacter cloaceae* DSM 30054, *Enterococcus faecalis* DSM 2570, *Escherichia coli* ATCC 25922, *Escherichia coli* ATCC 700414, *Escherichia coli* DSM 105393, *Candida albicans* ATCC 10231, *Candida glabrata* ATCC 90030, *Candida krusei* ATCC 14243, *Candida parapsilosis* ATCC 90018, *Candida tropicalis* ATCC 13803, *Gardnerella vaginalis* ATCC 14019, *Gardnerella vaginalis* ATCC 14018, *Listeria monocytogenes* DSM 12464, *Proteus mirabilis* DSM 30116, *Pseudomonas aeruginosa* DSM 1117, *Pseudomonas aeruginosa* DSM 3227, *Pseudomonas monteilii* ATCC 700476, *Staphylococcus aureus* ATCC 6538, and *Streptococcus agalactiae* DSM 2134 were used in the antagonistic assay. They were selected because their presence is often associated with different urogenital tract dysbioses. Each pathogen was cultured using the media and following the conditions suggested by ATCC or DSM. The antagonistic activity of the *L. rhamnosus* CA15 (DSM 33960) strain against pathogens was evaluated by the agar spot test as previously described by Pino et al. [15]. The antagonistic activity was ranked as absent (−, no inhibition zone); low (+, inhibition zone < 10 mm); intermediate (++, inhibition zone between 11 and 20 mm); and high (+++, inhibition zone > 20 mm). In order to identify the nature of the antimicrobial compounds, 10 µL of the CFS previously treated for pH, heat, catalase, and proteolytic enzymatic, as reported by Scillato and coworkers [31], was used for the antagonistic assay against the aforementioned pathogen stains. All experiments were carried out in triplicate.

#### 2.4.2. Hydrogen Peroxide, Organic Acid, and Lactic Acid Production

The ability to produce hydrogen peroxide (H_2_O_2_) was evaluated following the protocol reported by Pino et al. [15]. Based on the time required for the appearance of a blue coloration, the tested strain was scored as a nonproducer (absence of blue coloration), low (score 1, time > 20 min), medium (score 2, time 10–20 min), or high (score 3, time < 10 min) H_2_O_2_ producer.

Organic acids were quantified by HPLC following the protocol suggested by Chenoll and coworkers [32]. The assay was performed in triplicate, and the results are expressed as mean and standard deviation.

#### 2.4.3. In Vitro Antioxidant Activity

The antioxidant activity, explicated by the *L. rhamnosus* CA15 (DSM 33960) strain, was tested following the 2,2′-azino-di-[3-ethylbenzthiazoline sulfonate] (ABTS), 2,2-diphenyl-1-picrylhydrazyl hydrate (DPPH), and Superoxide Dismutase (SOD)-like assays. The scavenging activity toward radical cation 2,2′-azino-di-[3-ethylbenzthiazoline sulfonate] (ABTS) protocol was performed as reported by Pino et al. [28]. The free radical-scavenging capacity of different concentrations of the tested strain (3 mg/mL; 2.25 mg/mL; 1.5 mg/mL; 0.75 mg/mL) was measured by 2,2-diphenyl-1-picrylhydrazyl hydrate (DPPH)-free radical method as previously reported [33]. The SOD-like activity was measured using the pyrogallol autoxidation system as described by Marklund et al. [34]. Briefly, 50 µL of a pyrogallol solution (12 mM, in 1 mM HCl) was rapidly added to a 3 mL mixed solution system (0.05 M Tris–HCl buffer, 1 mM EDTA, and pH 8.2). After samples were thoroughly mixed, the absorbance at 325 nm was measured every 30 s for 5 min at 25 °C. According to the procedure described above, the assays were carried out in the presence of different concentrations of CA15 (DSM 33960) (0.75–3 mg/mL), expressed as mg of protein/mL of filtrate. Proteins were determined by the Nanodrop Protein quantification using a NanoDrop UV-Vis spectrophotometer (Thermo Scientific, Waltham, MA, USA). The autoxidation rate of pyrogallol (control) was determined from the slope of the absorbance curve. The change in absorbance in the presence of the antioxidant was compared with that of the control, and SOD-like activity was expressed according to the equation:SOD-like activity (%) = (A − B/A) × 100
where A and B are autoxidation rates of pyrogallol in the absence and presence of an antioxidant, respectively. Results are expressed as percentage of inhibition rate ± SEM (Standard Error of the Mean).

#### 2.4.4. Caco-2 Cell Cultures

Caco-2 cells were used as a stable in vitro model for the intestinal epithelium. Caco-2 cells (heterogeneous human epithelial colorectal adenocarcinoma cells) were purchased from American Type Culture Collection (ATCC; Rockville, MD, USA) and grown in DMEM supplemented with 10% fetal bovine serum (FBS), 0.1% streptomycin-penicillin, 1% L-glutamine, and 1% nonessential amino acids. Caco-2 cells were seeded at a concentration of 1.5 × 10^4^ cells per well of a 96-well, flat-bottomed 200-μL microplate. Dose–response experiments were performed in cells incubated at 37 °C in a 5% CO_2_-humidified atmosphere and cultured for 24 h in the presence and absence of selected concentrations of CA15 (DSM 33960) (1.5 mg/mL). Cell viability was measured by MTT assay as previously reported [35]. To induce in vitro intestinal inflammation, Caco-2 cells (1.5 × 10^4^) were seeded in six-well plates and cultured 14–21 days to reach the confluence. An experimental inflammatory condition in Caco-2 cell monolayers was induced by the exposure for 24 h to lipopolysaccharide (LPS) (100 μg/mL). A 24 h pretreatment with a selected concentration of CA15 (DSM 33960) (1.5 mg/mL) was applied before inflammatory stimuli.

#### 2.4.5. Mitochondrial Membrane Potential in Caco-2 Cell Line

Following a 24 h cotreatment of LPS (100 μg), cells were incubated with a 3 μM tetraethyl-benzimidazolyl-carbocyanine iodide (JC-1) staining solution (T3168, Invitrogen, Waltham, MA, USA) at 37 °C for 20 min and then washed with PBS in order to visualize the fluorescence with a fluorescence microscope (EVOS Fl AMG). The JC-1 probe aggregates formed a polymer in the mitochondrial matrix of healthy cells, producing a strong red fluorescence (Ex  = 585 nm, Em  =  590 nm). Otherwise, dysfunctional mitochondria presented JC-1 monomers, resulting in the emission of a green fluorescent signal (Ex  =  514 nm, Em =  529 nm). The results are expressed as the ratio of red/green fluorescence.

#### 2.4.6. Anti-Inflammatory Activity

The ability of the tested strain to exert anti-inflammatory activity was evaluated using the human promonocytic U937 cell line as reported by Pino et al. [28]. Gene expression of COX-1, COX-2, IL-8, and IL-10 was evaluated by quantitative real-time PCR (qRT-PCR) using the primer pairs and conditions previously described [28]. The analyses were performed in triplicate, and the results are reported as mean and standard deviation. In addition, conditioned media obtained from Caco-2 cells treated with LPS (100 µg/mL) and the combination with the selected concentration of CA15 (DSM 33960) (1.5 mg/mL) were used for quantitatively determining the IL-6 and IL-2 levels. eBioscience Instant ELISA kits (BMS213INST and BMS221INST, eBioscience, Vienna, Austria) were used following manufacturer’s instructions. The protein concentration was determined by plotting the OD reading against the standard curve. Results are expressed as pg/mL. Individual measurements were performed in triplicate, and data are reported as mean ± SEM.

#### 2.4.7. Adhesion to Caco-2 and Human Vaginal Epithelial Cells

The ability of the *L. rhamnosus* CA15 (DSM 33960) strain to adhere to Caco-2 (ATCC HTB-37) and human vaginal epithelial (VK2/E6E7 ATCC-CRL-2616) cell lines was evaluated as previously described [36]. Caco-2 cells were cultured as described above. The VK2/E6E7 cell line was grown in keratinocyte serum-free medium with 0.1 ng/mL human recombinant epidermal growth factor (EGF), 0.05 mg/mL bovine pituitary extract (BPE; Gibco), and added calcium chloride (0.4 mM; Alfa Aesar, Thermo Fisher, Tewksbury, MA, USA). The subculturing of VK2/E6E7 was carried out using Dulbecco’s modified Eagle’s medium and Ham’s F12 medium (DMEM:F-12; Gibco) containing 10% heat-inactivated fetal bovine serum. The adhesion assay was conducted in triplicate performing three separated experiments, and results are expressed as adhesion percentage (%).

#### 2.4.8. Adhesion to Intestinal Mucus

*L. rhamnosus* CA15 (DSM 33960) was grown overnight at 37 °C on MRS agar (Merck, Darmstadt, Germany). The strain was metabolically labeled by suspending one single colony in MRS broth supplemented with 10 µL/mL of [6-^3^H]-Thymidine. After overnight incubation at 37 °C, cells were harvested by centrifugation (2500× *g* for 10 min) and washed twice with phosphate-buffered saline (PBS; pH 7.2). The OD_600_ of the bacterial suspension was adjusted with PBS to 0.250 ± 0.02, which corresponded approximately to 10^8^ CFU/mL. Intestinal mucus was prepared by dissolving Mucin from porcine stomach, type III (Merck, Milan, Italy) in 1 M NaOH and diluting the solution with PBS in 1:20. A total of 75 ng of mucus was immobilized on polystyrene microtiter plate wells (Maxisorp, Nunc, Roskilde, Denmark) by overnight incubation at 4 °C. The mucus-coated microtiter plate wells were washed three times with 200 µL of PBS, and then 100 µL of cells was added to the wells and incubated at 37 °C for 1 h. Unattached bacteria were removed by washing the wells three times with 200 µL of PBS, whereas adhered cells were released and lyzed with 1% (*w/v*) sodium dodecyl sulfate (SDS) in 1 M NaOH (100 µL per well) by incubation at 37 °C overnight. The contents of the wells were transferred to microfuge tubes containing scintillation liquid (OptiPhase “HiSafe 3”, Wallac, Turku, Finland), and radioactivity was measured by liquid scintillation counter. Adhesion was expressed as the percentage of radioactivity recovered after adhesion relative to the radioactivity of the bacterial suspension added to the immobilized mucus. Adhesion was determined in three independent experiments, and each assay was performed in six replicates to calculate the intra-assay variation.

#### 2.4.9. Inhibition of Pathogen Adhesion and Displacement of Opportunistic Pathogens

*L. rhamnosus* CA15 (DSM 33960) was tested for its ability to displace and inhibit adhesion of *Streptococcus agalactiae* DSM 2134, *Enterobacter cloacae* DSM 30054, *Staphylococcus aureus* DSM 1104, *Candida glabrata* ATCC 90030, *Pseudomonas aeruginosa* DSM 1117, *Enterococcus faecalis* DSM 2570, *Gardnerella vaginalis* DSM 14018, *Candida parapsilosis* ATCC 90018, and *Candida albicans* ATCC10231. The strains were chosen based on their ability to adhere to mucus with percentages higher than 3% (data not shown). Except for *L. rhamnosus* DSM 33960, the strains were grown overnight using BHI (Sigma-Aldrich, St. Louis, MO, USA) and supplemented with 2% of Yeast Extract (Sigma-Aldrich, St. Louis, MO, USA). Adhesion inhibition and displacement were performed using mucus immobilized into polystyrene microtiter plate wells as above.

Adhesion inhibition was tested by adding unlabeled *L. rhamnosus* CA15 (DSM 33960) cells (100 µL, 10^8^ CFU/mL) to the mucus wells. After incubation at 37 °C for 1 h, unbound cells were removed by washing the wells three times with 200 µL of PBS after which radiolabeled pathogens (100 µL, 10^8^ CFU/mL) were added to the wells. After incubation at 37 °C for 1 h, unbound radiolabeled pathogens were removed by washing the wells three times with PBS, whereas the bound pathogens were released and lyzed with 1% (*w/v*) SDS in NaOH (0.1 mol/L) (100 µL per well) by incubation at 37 °C overnight. Radioactivity was assessed by liquid scintillation. Results are expressed as the percentage of radioactivity recovered after adhesion, relative to the radioactivity of the bacterial suspension added to the wells. The percentage of adhesion inhibition was calculated as the difference between the adhesion of the pathogen in the absence and presence of *L. rhamnosus* CA15 (DSM 33960) cells.

The ability of *L. rhamnosus* CA15 (DSM 33960) to displace adhered pathogens was evaluated by adding radiolabeled pathogens (100 µL, 10^8^ CFU/mL) to the mucus wells. After incubation at 37 °C for 1 h, unbound pathogens were removed by washing three times with PBS, and then unlabeled *L. rhamnosus* CA15 (DSM 33960) (100 µL, 10^8^ CFU/mL) was added to the wells. After incubation at 37 °C for 1 h, the wells were washed three times with PBS. After release and lysis of bound pathogens, the radioactivity was measured as described above. Displacement was calculated as the difference between the adhesion of pathogens before and after the addition of the *L. rhamnosus* CA15 (DSM 33960) cells. All assays were performed in three independent experiments, and each assay was performed in six replicates to calculate the intra-assay variation.

#### 2.4.10. Biofilm Formation Assay

The ability of the *L. rhamnosus* CA15 (DSM 33960) strain to produce biofilms was evaluated following the method proposed by dos Santos et al. [37] with slight modifications. An overnight microbial culture (10^9^ CFU/mL) was used as inoculum (5% *v/v*) in 200 μL of MRS broth medium with and without the addition of TWEEN 80 (1% *v/v*) and incubated at 37 °C, for 24 h and 72 h. For the assay, 96-well polystyrene microplates were used. After incubation, the wells were washed three times with PBS (pH 7.0), and 200 μL of 0.1% (*w/v*) crystal violet, prepared in an isopropanol–methanol–PBS (1:1:18 *v/v*) solution, was added and allowed to stain for 30 min. Excess dye was washed with sterile water and left to dry for 2–3 h. For the reading of the adhered biofilm inside the wells, 200 μL of 30% glacial acetic acid was added. The optical density (OD) was measured at 595 nm using an iMark™ Microplate Absorbance Reader (Biorad). A sterile medium was used as negative control (OD_c_). The strain was considered as nonbiofilm producers (OD ≤ OD_c_); weak biofilm producers (OD_c_ < OD ≤ 2 × OD_c_); moderate biofilm producers (2 × OD_c_ < OD ≤ 4 × OD_c_); strong biofilm producers (4 × OD_c_ < OD ≤ 8 × OD_c_), and very strong biofilm producers (8 × OD_c_ < OD).

#### 2.4.11. Determination of Antiadhesion Activity against *Candida* spp. Biofilm

The antiadhesion activity of the *L. rhamnosus* CA15 (DSM 33960) strain against *C. albicans* ATCC 10231, *C. glabrata* ATCC 90030, *C. krusei* ATCC 14243, *C. parapsilosis* ATCC 90018, and *C. tropicalis* ATCC 13803 was evaluated by precoating and coincubation experiments as reported by [38]. The analysis was performed in triplicate.

#### 2.4.12. Exopolysaccharide Production, Hydrophobicity, Autoaggregation, and Coaggregation Abilities

The phenol/sulfuric acid method was used to test the ability to produce exopolysaccharide (EPS). The amount (mg/L) of produced EPS was evaluated using glucose (50–500 mg/L) as standard [19,39].

Cell surface hydrophobicity (H%), autoaggregation (Auto-A%), and coaggregation (Co-A%) abilities were evaluated as described by Caggia et al. [40]. *E. coli* 555, *G. vaginalis* ATCC 14018, *G. vaginalis* ATCC 14019, *C. albicans* ATCC 10231, *C. glabrata* ATCC 90030, *C. krusei* ATCC 14243, *C. parapsilosis* ATCC 90018, and *C. tropicalis* ATCC 13803 were used as pathogenic strains. The analyses were performed in triplicate.

#### 2.4.13. Tolerance to Lysozyme, Low pH, and Bile Salts

Lysozyme tolerance was tested following the method previously described [19]. Tolerance to pH 2.0 and 3.0, as well as bile salt tolerance, was assessed as previously described [19]. For each assay, three independent experiments were performed using 9 log cfu/mL bacterial suspension of the *L. rhamnosus* CA15 (DSM 33960) strain previously cultured twice in MRS broth. Results are expressed as survival rate percentage (SR%) based on the initial and the final numbers of viable cells enumerated on MRS agar after incubation at 37 °C for 48 h. The analyses were performed in triplicate.

#### 2.4.14. Ability to Survive during Simulated Gastrointestinal Transit

The *L. rhamnosus* CA15 (DSM 33960) strain was tested for the ability to survive during the in vitro gastrointestinal transit. The survivability was evaluated on simulated gastric juice (SGJ, pH 2.5) and simulated intestinal fluid (SIF, pH 7.5) using 9 log cfu/mL bacterial suspension of the tested strain previously cultured twice in MRS broth, according to Pino et al. [19]. The assay was performed in triplicate, and results are expressed as survival rate percentage (SR%), based on the initial and the final numbers of viable cells. The analyses were performed in triplicate.

### 2.5. Statistical Analysis

All data are expressed as a mean and standard deviation or as frequencies (percentages) of triplicate independent experiments. Data related to RT-PCR of COX-1, COX-2, IL-8, and IL-10 gene-level expression were subjected to one-way ANOVA followed by Tukey’s multiple comparison test. Differences were considered statistically significant at *p*  ≤  0.05. All statistical analyses were performed using SPSS Version 25.0 (Armonk, NY, USA, IBM Corp.).

## 3. Results

### 3.1. Genetic Analysis of the L. rhamnosus CA15 (DSM 33960) Strain

The alignment of the nucleotide sequence of the 16S rRNA gene of the *L. rhamnosus* CA15 (DSM 33960) strain showed 100% identity (i.e., 613 nucleotides out of 613 were identical) with the 16S rRNA gene of the *L. rhamnosus*-type strain ATCC 7469 (sequence accession no. JQ580982.1). The CA15 (DSM 33960) strain was therefore identified as belonging to the species *Lacticaseibacillus rhamnosus* (old name *Lactobacillus rhamnosus)*. The genetic fingerprint via PFGE is shown in Figure 1.

### 3.2. Safety Evaluation

The *L. rhamnosus* CA15 (DSM 33960) strain was considered safe because it showed neither hemolysis halos after growth on blood agar plates nor zones of gelatin hydrolysis on gelatin agar plates overlaid with saturated ammonium sulfate solution. In addition, the CA15 (DSM 33960) strain did not show DNAse activity and mucin degradation ability.

The antibiotic susceptibility profile (MIC values) of the tested CA15 (DSM 33960) strain is shown in Table 1. Susceptibility to all the antibiotics suggested by the EFSA was detected, whereas high MIC values for Metronidazole (>600 µg/mL), Clotrimazole (>256 µg/mL), and Boric acid (>10.000 µg/mL) were observed. In addition, as reported in Table 1, no relevant amounts of the assayed biogenic amines were detected by quantification performed according to ISO19343:2017.

### 3.3. Antagonistic Activity against Pathogens

The results of the antimicrobial activity against pathogens displayed by the *L. rhamnosus* CA15 (DSM 33960) strain are reported in Table 2. Overall, a broad spectrum of antagonistic activity was detected except for *Enterobacter cloaceae* DSM 30054, *Enterococcus faecalis* DSM 2570, *Escherichia coli* ATCC 25922, *Escherichia coli* DSM 105393, and *Proteus mirabilis* DSM 30116. The highest antimicrobial activity was displayed against *Candida albicans* ATCC10231, *Candida krusei* ATCC 14243, *Gardnerella vaginalis* ATCC 14018, and *Streptococcus agalactiae* DSM 2134. Based on the CFS treatments, the antagonistic activity was, in all cases, attributed to organic acid production.

### 3.4. Hydrogen Peroxide, Organic Acid, and Lactic Acid Production

The *L. rhamnosus* CA15 (DSM 33960) strain, with a score of three, was classified as a high hydrogen peroxide producer because the appearance of an intense blue coloration was observed less than 10 min after plate air exposure (Table 3). Regarding organic acid production, a high concentration of lactic acid (799.29 mmol/L) was quantified. In addition, propionic (98.54 mmol/L), succinic (98.48 mmol/L), and butyric (39.61 mmol/L) and acetic (18.31 mmol/L) acids were detected (Table 3).

### 3.5. In Vitro Antioxidant and Anti-Inflammatory Activities and Adhesion Ability to Caco-2 and Human Vaginal Epithelial Cells

The antioxidant activity was evaluated in *cell-free* models by ABTS, DPPH Radical Scavenging, and SOD-like assays. Figure 2 shows the results of the ABTS test suggesting the ability of the *L. rhamnosus* CA15 (DSM 33960) strain to inhibit the peroxidation of the linoleic acid. Interestingly, no statistically significant difference (*p* > 0.05) was found in antioxidant activity between the tested strain and the well-known antioxidant compound α-tocopherol. DPPH Radical Scavenging and SOD-like assays were performed using different concentrations (3, 2.25, 1.5, and 0.75 mg/mL) of the filtered broth, obtained after the growth of the *L. rhamnosus* CA15 (DSM 33960) strain, and the results are displayed in Figure 3A,B). Overall, similar DPPH radical scavenging activity, with a percentage of about 80%, was detected using 3, 2.25, and 1.5 mg/mL of the *L. rhamnosus* CA15 (DSM 33960) strain culture filtrate (Figure 3A). On the contrary, the tested strain was able to scavenge superoxide anion in a dose-dependent manner, reaching about 80% at a concentration of 3 mg/mL (Figure 3B).

The ability of LPS to induce oxidative stress in the Caco-2 cell line was evaluated by measuring the levels of JC-1, a cationic dye that accumulates in the mitochondria, as dimer JC-1 or monomer JC-1. As shown in Figure 3C, a low signal of JC-1 dimer (red fluorescence) and a high presence of JC-1 monomer (green fluorescence) were detected in LPS-treated Caco-2 cells, indicating the induction of oxidative damage and mitochondrial dysfunction. In addition, as shown in Figure 3D, the LPS treatment was able to induce oxidative stress in the Cac-2 cell line with slight cytotoxicity. No significant differences were observed in the CA15 (DSM 33960) cells treated and in the CA15 (DMS 33960)–LPS-treated cells, compared to the control group.

A significant reduction in the IL-6 levels was detected in LPS-treated Caco-2 cell lines after treatment with the *L. rhamnosus* CA15 (DSM 33960) strain culture filtrate, suggesting the ability of the tested strains to reduce both oxidative stress and inflammation (Figure 3E). On the contrary, the treatment of LPS-treated Caco-2 cell lines with the *L. rhamnosus* CA15 (DSM 33960) strain culture filtrate was able to induce an increase in the IL-2 levels (Figure 3F).

As reported in Table 4, the *L. rhamnosus* CA15 (DSM 33960) strain showed anti-inflammatory activity in differentiated human macrophages treated with LPS. In detail, compared to the untreated cells, the treatment with the *L. rhamnosus* CA15 (DSM 33960) strain culture filtrate determined the downregulation of both COX-2 and IL-8 genes, whereas the IL-10 gene was upregulated. In addition, the tested strain exhibited binding ability to both Caco-2 and VK2/E6E7 cell lines, with rates of 36.15% ± 0.32 and 34.71% ± 0.09, respectively (Table 4).

Antioxidant activity (Abs at 500 nm) was measured under a linoleic acid oxidation system for 8 days. α-Tocopherol (1 mg/mL) was used as a positive control. Data are reported as means of three independent experiments.

### 3.6. Adhesion to Intestinal Mucus and Ability to Inhibit the Pathogen Adhesion and to Displace Pathogens

Table 5 shows the results of the adhesion ability of the *L. rhamnosus* CA15 (DSM 33960) strain to intestinal mucus as well as the results of the ability to displace and inhibit the adhesion of selected potentially pathogen strains. In detail, the *L. rhamnosus* CA15 (DSM 33960) strain showed the ability to adhere to intestinal mucus with a percentage equal to 11.99% (Table 5). In addition, the *L. rhamnosus* CA15 (DSM 33960) strain showed the ability to inhibit the adhesion as well as to displace all the tested pathogens with percentages higher than 50% (Table 5).

The results are expressed in percentage as means and standard deviation of the three independent experiments.

### 3.7. Biofilm Formation

The results of the biofilm formation are reported in Table 6. In detail, the tested strain exhibited strong biofilm production in MRS after both 24 and 72 h of incubation, whereas when using MRS with Tween 80, a very strong biofilm production was recorded after 72 h of incubation (Table 6).

### 3.8. Antiadhesion Activity against Candida spp. Biofilm

The results of the antiadhesion activity against *C. albicans* ATCC 10231, *C. glabrata* ATCC 90030, *C. krusei* ATCC 14243, *C. parapsilosis* ATCC 90018, and *C. tropicalis* ATCC 13803 are shown in Table 7. Overall, the antiadhesion activity was displayed by the *L. rhamnosus* CA15 (DSM 33960) strain in both the precoating and coincubation assays. The antiadhesion percentages ranged from 87% to 50.1% and from 48.4% to 20.1% in the precoating and coincubation assays, respectively. In detail, in the precoating assay, antiadhesion activity was shown against *C. glabrata* (87%) and *C. albicans* (84.6%), followed by *C. parapsilosis* (75.7%), *C. tropicalis* (71.6%), and *C. krusei* (50.1%). In the coincubation test, the highest antiadhesion activity was displayed against *C. tropicalis*, with a 48.4% reduction in biofilm formation, whereas the lowest was detected against *C. albicans* (20.1%) (Table 7).

### 3.9. Exopolysaccharide Production, Hydrophobicity, Autoaggregation, and Coaggregation Abilities

The results of the exopolysaccharide (EPS) production and surface properties shown by the *L. rhamnosus* CA15 (DSM 33960) strain are reported in Table 8. The tested strain was able to produce 292 mg/L of EPS. Both cell surface hydrophobicity (86.13%) and autoaggregation properties (71.80%) were detected. In addition, the CA15 (DSM 33960) strain was able to coaggregate with all the tested pathogens. The highest coaggregation percentage was observed in the presence of *C. albicans* ATCC 10231 (86.12%), whereas the lowest value was observed in the case of coaggregation with *E. coli* 555 (64.18%) (Table 8).

### 3.10. Survival under In Vitro Gastrointestinal Conditions

The tolerance to lysozyme, acidic conditions, and bile salts as well as the ability to survive during simulated gastrointestinal (GI) transit is reported in Table 9. Overall, the *L. rhamnosus* CA15 (DSM 33960) strain showed the ability to survive under the tested stressful conditions with a survival rate (SR%) higher than 80%. In detail, the *L. rhamnosus* CA15 (DSM 33960) strain was able to survive under acidic conditions (pH 2.0 and 3.0) after both 2 and 4 h of incubation. Regarding the tolerance to lysozyme, the CA15 (DSM 33960) strain was classified as lysozyme-resistant [41] because it showed a survival rate of 95% and 93% after 30 and 120 min of incubation, respectively. The tolerance to both 0.5% and 1% of bovine bile salts, with an SR% higher than 80%, was recorded after both 2 h and 4 h of incubation (Table 9). A survival rate higher than 90% was observed after treatment with simulated gastric juice (SGJ) and simulated intestinal fluid (SIF). In fact, starting from an initial number of viable cells of 9.13 log cfu/mL, the *L. rhamnosus* CA15 (DSM 33960) strain exhibited several viable cells of 9.10 log cfu/mL after SGJ treatment (SR% of 99.7%) and 8.74 log cfu/mL after SIF treatment (SR% of 95.7%) (Table 9).

## 4. Discussion

It is well known that, during reproductive age, a balanced vaginal microbiota is dominated by lactobacilli, while their reduction, and the increase in pathogens, is often associated with a dysbiotic state [42]. Although antimicrobial drugs have been traditionally used to treat dysbiosis, several side effects, such as short-term relapse and the promotion of antibiotic resistance, are frequently reported. For these reasons, alternative therapeutic strategies, able to modulate the microbiota and correct the imbalance, from an ecological approach, have been investigated. In this context, the probiotic supplementation, alone or complementary to antibiotic therapy, has gained increased attention to restore vaginal health avoiding the recurrence of bacterial vaginosis (BV) and vulvovaginal candidiasis (VVC) [43,44]. Although several probiotic strains have been characterized, only few are host-specific. It has been suggested that probiotic vaginal isolates, when administered, have increased colonization and adaptation abilities to the vaginal niche. The workflow of the present study was designed to evaluate the potential probiotic properties of the vaginal isolate *L. rhamnosus* CA15 (DSM 33960) strain as a potential candidate for women’s health.

Noteworthy, probiotics must satisfy safety requirements. In this context, the inability to produce hemolysins and gelatinases, as well as to exert mucin degradation and DNase activities, suggest that the *L. rhamnosus* CA15 (DSM 33960) strain can be safely used. In addition, regarding the horizontal transfer of genes encoding antibiotic resistance [45,46], no transferable antibiotic resistance should be carried by promising probiotic strains. In the present study, the *L. rhamnosus* CA15 (DSM 33960) strain did not show resistance to the antibiotics suggested by the EFSA, whereas resistance to metronidazole, clotrimazole, and boric acid was detected. The high resistance to antibiotics routinely used to treat BV and VVC suggests the possible use of the CA15 (DSM 33960) strain as an adjuvant to antibiotic treatment, ensuring survival during antibiotic administration.

The ability of probiotics to counteract pathogen invasion and colonization is a prerequisite to modulate the vaginal microbiota in a homeostatic way. Antimicrobial substances, such as hydrogen peroxide (H_2_O_2_) and organic acids, produced by probiotic bacteria, play a key role in the maintenance of a healthy vaginal ecosystem. Although hydrogen peroxide is able to exert a microbicide effect, there is no consensus about its activity against pathogens due to both a low physiological concentration in the vagina and the neutralizing effects of semen and vaginal fluid [47]. It is well established that lactic acid, by acidifying the vagina, acts as a virucide and microbicide along with an immunomodulatory agent [48]. In the present study, we observed that the tested strain was able to produce both H_2_O_2_ and lactic acid. In particular, the amount of lactic acid produced could be related to the broad antagonistic effect displayed by the CA15 (DSM 33960) strain against the tested pathogens [48]. The absence of antimicrobial activity after CFS neutralization suggests that the acid environment had a substantial action against the growth of the tested pathogens. However, other components, not yet identified, may occur in CFS. Further study based on the metabolomic approach will be carried out to characterize in depth the components of CFS and to evaluate their synergistic action to inhibit pathogens [49,50].

Along with the production of metabolites with antimicrobial properties, probiotic strains should be able to colonize and survive in harsh environments. The ability to adhere to the intestinal mucosa is considered a prerequisite for probiotics, allowing the colonization, albeit transient, of the human intestinal tract [9,14]. In vitro experiments showed the ability of the CA15 (DSM 33960) strain to adhere to intestinal mucus as well as to both Caco-2 and VK2/E6E7 cell lines with adhesion percentages higher than those previously obtained by *L. rhamnosus* GG [15]. In addition, the tested strain showed hydrophobicity and the ability to autoaggregate and coaggregate with pathogens. A higher hydrophobicity and autoaggregation ability was displayed by the CA15 strain compared to *L. rhamnosus* GG [40]. It is well known that the ability to adhere is related to the predisposition to self-aggregate, which, in turn, can exert a protective role by inhibiting the growth and adhesion of pathogens [14]. Moreover, the exclusion/competition behavior of probiotics is also related to coaggregation with pathogens [37]. According to that, the *L. rhamnosus* CA15 (DSM 33960) strain showed a significant capability to coaggregate with all the tested pathogens as well as to displace and inhibit their adhesion.

Recently, several health benefits (e.g., immunomodulation, antioxidant, antiviral, and antiyeast properties) have been associated with exopolysaccharide (EPS) produced by lactobacilli [51]. According to that, the ability of the CA15 (DSM 33960) strain to produce EPS could be useful to maintain vaginal health. Other interesting properties of the *L. rhamnosus* CA15 (DSM 33960) strain are the antioxidant and anti-inflammatory profiles. The ability to counteract the lipid peroxidation, and in turn oxidative stress, is considered central for the prevention of tissue damage and the development of human pathologies. In particular, the superoxide radical scavenging activity is an important enzymatic antioxidant defense mechanism catalyzed by the enzyme superoxide dismutase (SOD), which catalyzes the dismutation of the superoxide anion into oxygen and hydrogen peroxide to protect the body from oxygen toxicity. In this study, we measured the scavenging activity of the superoxide radical generated by the self-oxidation of pyrogallol. In addition, the anti-inflammatory profile suggests a potential role of the CA15 (DSM 33960) strain to decrease the levels of proinflammatory cytokines and maintain the inflammatory homeostasis by increasing the levels of the anti-inflammatory molecules. In particular, the tested strain was able to induce a decrease in the IL-6 and, in turn, determine an increase in the IL-2 in LPS-treated Caco-2 cells. It is well known that IL-6 is an important proinflammatory cytokine involved in the development of inflammatory bowel diseases (IBD) [52,53], whereas considerable evidence suggest the important role played by IL-2 in maintaining a healthy immune response in the gut. In fact, a high rate of colitis, with striking clinical and morphological similarities to ulcerative colitis, was reported in a IL-2-deficient mouse model [54]. In addition, low doses of IL-2 are used for the treatment of moderate to severe ulcerative colitis [55,56].

Recently, the ascending colonization hypothesis was proposed suggesting that probiotics orally delivered can ascend to the vagina and colonize them after excretion from the rectum [2]. To make this possible, probiotic strains must survive through the GI tract. Interestingly, the CA15 (DSM 33960) strain, tested in the present study, exhibited a high survival rate at a low pH, in the presence of lysozymes and bile salts as well as during the in vitro-simulated GI digestion. In particular, the survival percentages detected after exposure to SGJ and SIF were higher than those displayed by the *L. rhamnosus* GG [40].

Hence, the probiotic potential of the *L. rhamnosus* CA15 (DSM 33960) strain supports its use as a therapeutic strategy for women’s health. Further studies will be designed to set up an in vitro coculture model using Caco-2 cell lines and U937 cells to investigate the crosstalk between intestinal cells and the cells of the immune system treated with the *L. rhamnosus* CA15 (DSM 33960) strain. In addition, the in vitro therapeutic potential of the *L. rhamnosus* CA15 (DSM 33960) strain will be validated in a cohort of patients with vaginal dysbiosis.

## 5. Patents

Data reported in this manuscript were submitted to the Italian Ministero dello Sviluppo Economico to be recognized as patent for industrial invention (submission no. 102022000016542).

## Figures and Tables

**Figure 1 nutrients-14-04902-f001:**
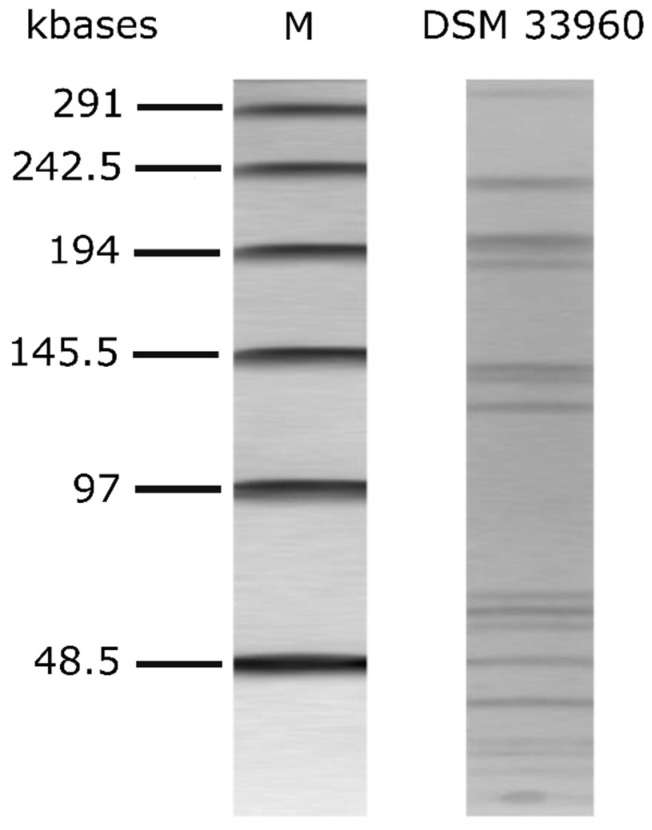
PFGE profiles of *Asc*I-digested genomic DNA of *L. rhamnsous* CA15 (DSM 33960) strain. M: Lambda PFG marker (New England Biolabs, Inc., Ipswich, MA, USA).

**Figure 2 nutrients-14-04902-f002:**
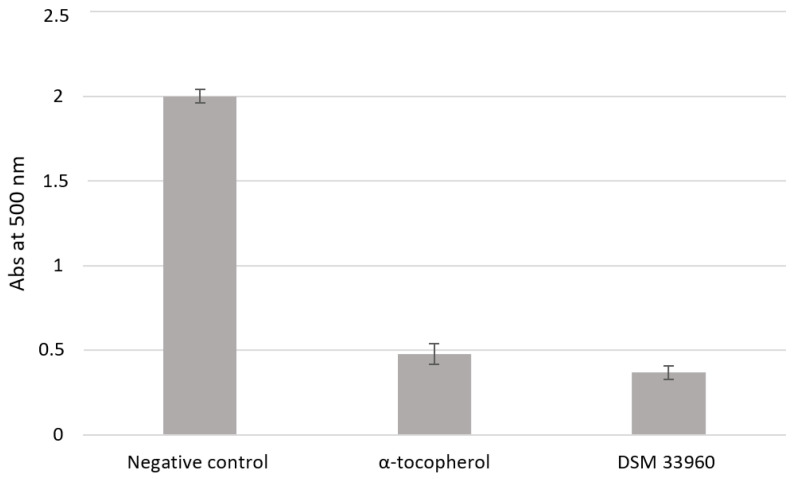
Lipid peroxidation inhibitory activity of the *L. rhamnsous* CA15 (DSM 33960) strain.

**Figure 3 nutrients-14-04902-f003:**
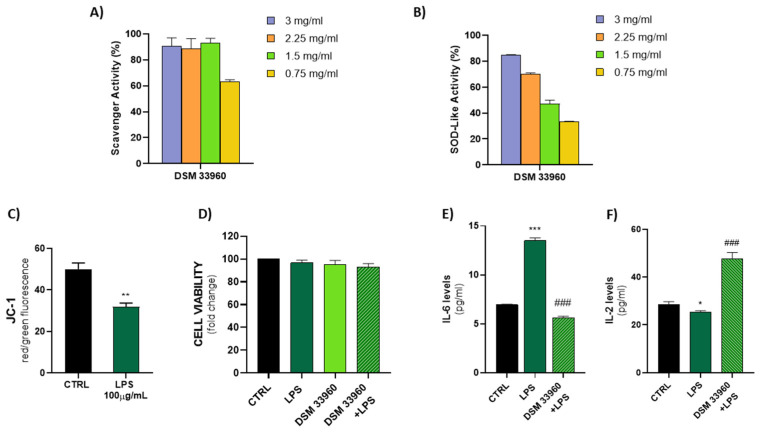
DPPH scavenger and SOD-like activity explicated by the CA15 (DSM 33960) strain tested at different concentrations (3, 2.25, 1.5, and 0.75 mg/mL) (Panels **A**,**B**); LPS treatment (100 µg/mL) effect on mitochondrial membrane potential (Panel **C**) (** *p* < 0.005 vs. CTRL); Assessment of Caco-2 cell viability after treatments (Panel **D**); Measurement of IL-6 and IL-2 levels in cell supernatants by ELISA (Panels **E**,**F**) (* *p* < 0.05, *** *p* < 0.0005 vs. CTRL; and ^###^
*p* < 0.0005 vs. LPS).

**Table 1 nutrients-14-04902-t001:** Safety properties of the *L. rhamnosus* CA15 (DSM 33960) strain.

	MIC (µg/mL) ^a^	
Tested Antibiotics	CA15 (DSM 33960)	Breakpoints ^b^
Ampicillin	2	4
Chloramphenicol	1	4
Clindamycin	0.25	1
Erythromycin	0.5	1
Gentamicin	8	16
Kanamycin	32	64
Streptomycin	16	32
Tetracycline	4	8
Boric acid	>10,000	nr
Clotrimazole	>256	nr
Metronidazole	>600	nr
Biogenic amines (g/L)		
Histamine	nd	
Spermine	nd	
Tyramine	0.003	
Putrescine	nd	
Cadaverine	nd	

^a^ MICs determined by microdilution method; ^b^ Breakpoints proposed by the European Food Safety Authority (EFSA, 2012); nr: not required in the case of *L. rhamnosus* according to EFSA; nd: not detected.

**Table 2 nutrients-14-04902-t002:** Antimicrobial activity against pathogens exhibited by the *L. rhamnosus* CA15 (DSM 33960) strain.

Antimicrobial Activity	
Tested Pathogens	CA15 (DSM 33960) Strain
*Enterobacter cloaceae* DSM 30054	−
*Enterococcus faecalis* DSM 2570	−
*Escherichia coli* ATCC 25922	−
*Escherichia coli* ATCC 700414	+
*Escherichia coli* DSM 105393	−
*Candida albicans* ATCC 10231	+++
*Candida glabrata* ATCC 90030	++
*Candida krusei* ATCC 14243	+++
*Candida parapsilosis* ATCC 90018	+
*Candida tropicalis* ATCC 13803	++
*Gardnerella vaginalis* ATCC 14019	++
*Gardnerella vaginalis* ATCC 14018	+++
*Listeria monocytogenes* DSM 12464	++
*Proteus mirabilis* DSM 30116	−
*Pseudomonas aeruginosa* DSM 1117	+
*Pseudomonas aeruginosa* DSM 3227	+
*Pseudomonas monteilii* ATCC 700476	++
*Staphylococcus aureus* ATCC 6538	++
*Streptococcus agalactiae* DSM 2134	+++

(−), no inhibition zone; (+), inhibition zone < 10 mm; (++), inhibition zone between 11 and 20 mm; (+++), inhibition zone > 20 mm.

**Table 3 nutrients-14-04902-t003:** Hydrogen peroxide (H_2_O_2_) production and organic acid profile exhibited by the *L. rhamnosus* CA15 (DSM 33960) strain.

	CA15 (DSM 33960)
H_2_O_2_	3
Organic acids (mmol/L)	
Acetic acid	18.31
Butyric acid	39.61
Lactic acid	799.29
Propionic acid	98.54
Succinic acid	98.48

**Table 4 nutrients-14-04902-t004:** Antioxidant, anti-inflammatory, and adhesion to Caco-2 and VK2/E6E7 cell lines exhibited by the *L. rhamnosus* CA15 (DSM 33960) strain.

Anti-Inflammatory Activity on U937 Cells					Adhesion %	
	COX-1	COX-2	IL-8	IL-10	Caco-2	VK2/E6E7
LPS	2.13 ± 0.11	15.29 ± 0.73	4.16 ± 0.05	128.90 ± 1.28		
CA15 (DSM 33960)	2.19 ± 0.26	0.09 ± 0.07 *	0.71 ± 0.08 *	524.71 ± 1.05 *	36.15 ± 0.32	34.71 ± 0.09

COX-1, COX-2, IL-8, and IL-10 data are expressed as increased folds compared to untreated cells and standard deviation; * *p*  <  0.05 with respect to lipopolysaccharide (LPS) treatment.

**Table 5 nutrients-14-04902-t005:** Adhesion of the *L. rhamnosus* CA15 (DSM 33960) strain to intestinal mucus and effect on adhesion ability of pathogens.

Tested Strains	Adhesion	Displacement	Exclusion
*L. rhamnosus* CA15 (DSM 33960)	11.99 ± 0.27		
*Streptococcus agalactiae* DSM 2134		73.11 ± 0.16	87.41 ± 0.08
*Enterobacter cloacae* DSM 30054		52.00 ± 0.14	82.00 ± 0.26
*Staphylococcus aureus* DSM 1104		51.56 ± 0.11	81.58 ± 0.15
*Candida glabrata* ATCC 90030		92.79 ± 0.14	53.44 ± 0.40
*Pseudomonas aeruginosa* DSM 1117		62.12 ± 0.21	71.15 ± 0.19
*Enterococcus faecalis* DSM 2570		52.33 ± 0.24	62.68 ± 0.34
*Gardnerella vaginalis* DSM 14018		54.86 ± 0.56	67.43 ± 0.08
*Candida parapsilosis* ATCC 90018		77.07 ± 0.19	86.24 ± 0.15
*Candida albicans* ATCC10231		63.27 ± 1.02	91.53 ± 1.19

**Table 6 nutrients-14-04902-t006:** Ability of the *L. rhamnosus* CA15 (DSM 33960) strain to produce biofilm.

Biofilm Production		
	24 h	72 h
MRS	strong	strong
MRS with Tween 80	strong	very strong

**Table 7 nutrients-14-04902-t007:** Percentage values of antiadhesion activity of CFS against *Candida* spp.

Antiadhesion Activity of CFS against Biofilm *Candida* Species		
	Precoating	Coincubation
*C. albicans*	84.6 ± 0.14	20.1 ± 0.70
*C.tropicalis*	71.6 ± 0.09	48.4 ± 0.60
*C. glabrata*	87.0 ± 0.2	29.6 ± 0.39
*C. krusei*	50.1 ± 0.17	45.9 ± 0.70
*C. parapsilosis*	75.7 ± 0.2	23.9 ± 0.88

**Table 8 nutrients-14-04902-t008:** Exopolysaccharide (EPS) production, hydrophobicity, autoaggregation, and coaggregation abilities exhibited by the *L. rhamnosus* CA15 (DSM 33960) strain.

	CA15 (DSM 33960)
EPS (mg/L)	292 ± 2.13
H% ^a^	86.13 ± 0.05
Auto-A% ^a^	71.80 ± 0.07
CoA% ^a^	
*E. coli 555*	64.18 ± 0.16
*G. vaginalis* ATCC 14018	71.23 ± 0.11
*G. vaginalis* ATCC 14019	68.18 ± 0.33
*C. albicans* ATCC10231	86.12 ± 0.17
*C. glabrata* ATCC 90030	82.63 ± 0.18
*C. krusei* ATCC 14243	85.14 ± 0.11
*C. parapsilosis* ATCC 90018	79.21 ± 0.21
*C. tropicalis* ATCC 13803	73.15 ± 0.13

^a^ Results of surface properties (H%: hydrophobicity; Auto-A%: autoaggregation; and CoA%: coaggregation) are expressed as average percentage values and standard deviations of three separate experiments.

**Table 9 nutrients-14-04902-t009:** Survivability of the *L. rhamnsous* CA15 (DSM 33960) strain in presence of lysozyme, acidic conditions, and bile salts and during in vitro-simulated gastrointestinal transit.

Tested Activity	Condition	Time	Survival Rate (SR%)
Lysozyme tolerance		30 min	95%
		120 min	93%
Tolerance to low pH	3.0	2 h	92%
		4 h	86%
	2.0	2 h	89%
		4 h	83%
Tolerance to bile salts	0.5%	2 h	91%
		4 h	87%
	1%	2 h	89%
		4 h	82%
Simulated GI digestion	SGJ *		99.7%
	SIF **		95.7%

* Simulated gastric juice; ** Simulated intestinal fluid. Survival rate (SR%) was calculated based on the initial and the final numbers of viable cells enumerated on MRS agar after 48 h.

## Data Availability

The datasets generated and analyzed during the current study are available from the corresponding authors upon reasonable request.

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
