# Peer review of "Lacticaseibacillus rhamnosus CA15 (DSM 33960) as a Candidate Probiotic Strain for Human Health"

_nutrients, 2022, doi:10.3390/nu14224902_

Round 1

Reviewer 1 Report

It is a very interesting and complete research, however there are some details to be corrected for a better presentation.

p2 L47-48. Change to "...of lactobacilli have a GRAS (generally recognized as safe) status and...

p2 L65. Delete "female"

p2 L79. Please provide a reference for the genetic identification.

p3 L127-143. Please use italic fonts for all scientific names and the word in vitro.

p4 L150. Delete the word "chromatography"

p4 L175-176. Define "SEM"

p7 L312. Please indicate the concentration and type of bile salts used.

p7 L320-321. Please indicate the pH value of SGJ and SIF.

p7 L327. Please indicate the number of replicas of the experiments.

p10 L379. Please write "peroxide" instead of "peroxidase"

p10 L389. If p<0.05 then there are statistically significant differences. Please correct.

p14 L503. Please compare the probiotic properties of L. rhamnosus CA15 with those of L. rhamnosus GG.

Author Response

p2 L47-48. Change to "...of lactobacilli have a GRAS (generally recognized as safe) status and...

Thanks for your suggestion. We modified the sentence as you suggested.

p2 L65. Delete "female"

Thanks for your suggestion. We deleted "female"

p2 L79. Please provide a reference for the genetic identification.

Thanks for your suggestion. We reported references about genetic identification.

p3 L127-143. Please use italic fonts for all scientific names and the word in vitro.

Thanks for your suggestion. We used italic fonts for all scientific names and the word in vitro

p4 L150. Delete the word "chromatography"

Thanks for your suggestion. We deleted the word "chromatography"

p4 L175-176. Define "SEM"

Thanks for your suggestion. We added the definition of SEM (Standard Error of the Mean)

p7 L312. Please indicate the concentration and type of bile salts used.

Thanks for your comment. Commercial bile salts, extracted from purified fresh bile, at both 0.5% and 1% were used to test the ability to survive in presence of bile salts.  

p7 L320-321. Please indicate the pH value of SGJ and SIF.

Thanks for your suggestion. We specified in the text the pH of both SGJ and SIF

p7 L327. Please indicate the number of replicas of the experiments.

Thanks for your suggestion. We specified in the text that the analysis was performed in triplicate.

p10 L379. Please write "peroxide" instead of "peroxidase"

Thanks for your suggestion. We changed the word as you suggested

p10 L389. If p<0.05 then there are statistically significant differences. Please correct.

Thanks for your comment and sorry for the mistake about the used symbol. In real, the statistical analysis did not reveal any significant difference in antioxidant activity between the tested strain and the well-known antioxidant compound α-tocopherol. So we put in the text the symbol “>” instead of “<”   

p14 L503. Please compare the probiotic properties of L. rhamnosus CA15 with those of L. rhamnosus GG.

Thanks for your suggestion. For some of the tested probiotic features, we included, in the discussion section, a comparison with L. rhamnosus GG.

Reviewer 2 Report

The manuscript of Pino et al. has an interesting topic: the isolation and characterization of a new probiotic strain, L. rhamnosus CA15 to be used for potential applications in human health. The authors performed many experiments for a deep and exhaustive characterization of the new strain. Although the data obtained are interesting it is necessary to clarify some aspects as suggested in order to improve the manuscript. In conclusion, a major revision is suggested.   General Comments: 1.     In my opinion the authors should remove the terms “pharmaceutical” and “alternative to antibiotics” (pg. 2). I’m sure that probiotics and their products are active and probably, new molecules will be isolated by the Cell Free Supernatant in future, but, actually, they are considered food supplements and cannot replace antibiotics for the eradication of microbial infections. I certainly agree with the authors in their use to prevent infections and colonization by pathogens (Introduction).

The introduction should be improved. In particular, more details should be inserted regarding the lactobacilli involved in the prevention of vaginal dysbiosis focusing the attention on L. rhamnosus and the importance and role of biofilm development.

  1. The authors should write a sentence concerning the choice of pathogens/opportunistic pathogens used in the study (methods).
  2. The microbial genus and species should be written in italics. Please check the typing errors in the text.
  3. The isolation of the strain from a patient requires the approval of an Ethics Committee. Please, add the project indentification code.
  4. Regarding the antimicrobial activity of L. rhamnosus CFS, 50 microliters might be an amount toxic for human cell lines. Please, perform a citotoxicity evaluation on human vaginal epithelial cell (methods and results).
  5. Please comments the antimicrobial activity of CFS in the discussion.
  6. In my opinion the antagonist activity of CFS cannot be associated only with organic acids, you cannot exclude a synergistic activity of several components inside the CFS. A metabolomic approach might be useful to obtain more information for future studies.
  7. Please check the references accurately, according to the journal guidelines

Author Response

General Comments: 

1.     In my opinion the authors should remove the terms “pharmaceutical” and “alternative to antibiotics” (pg. 2). I’m sure that probiotics and their products are active and probably, new molecules will be isolated by the Cell Free Supernatant in future, but, actually, they are considered food supplements and cannot replace antibiotics for the eradication of microbial infections. I certainly agree with the authors in their use to prevent infections and colonization by pathogens (Introduction).

Thanks for your comment. We rephrased the sentence according to your suggestion.

The introduction should be improved. In particular, more details should be inserted regarding the lactobacilli involved in the prevention of vaginal dysbiosis focusing the attention on L. rhamnosus and the importance and role of biofilm development.

Thanks for your comment. We improved the introduction section by reporting recently published evidence about the usefulness of L. rhamnosus strains to balance the vaginal microbiota.

  1. The authors should write a sentence concerning the choice of pathogens/opportunistic pathogens used in the study (methods).

Thanks for your comment. We justified in the text the choice pathogens/opportunistic pathogens used in the study.

  1. The microbial genus and species should be written in italics. Please check the typing errors in the text.

Thanks for your comment. All genera and species are written in italics.

  1. The isolation of the strain from a patient requires the approval of an Ethics Committee. Please, add the project indentification code.

Thanks for your comment. We specified in the text the approval number provided by the Local Ethical Committee

  1. Regarding the antimicrobial activity of L. rhamnosus CFS, 50 microliters might be an amount toxic for human cell lines. Please, perform a citotoxicity evaluation on human vaginal epithelial cell (methods and results).

Thanks for your comment. As reported in the text, the potential toxicity to human cells was evaluated using CaCo-2 cell line. Results showed the absence of cytotoxicity. Further studies are ongoing to exclude the possible cytotoxic effect on other cell lines such as human vaginal epithelial cells.

  1. Please comments the antimicrobial activity of CFS in the discussion.

Thanks for your comment. We commented the antimicrobial activity of CFS in the discussion section 

  1. In my opinion the antagonist activity of CFS cannot be associated only with organic acids, you cannot exclude a synergistic activity of several components inside the CFS. A metabolomic approach might be useful to obtain more information for future studies.

Thanks for your comment and for the suggestion for further investigations. Based on the different treatments applied to CFS, authors postulated the involvement of organic acids in the observed antagonistic activity although the possible implication of several molecules, along with organic acids, could be possible. Further studies will be conducted, from a metabolomic point of view, in order to in-depth investigate the type and amount of metabolites produced and their possible correlation with the documented antimicrobial activity against pathogens.     

  1. Please check the references accurately, according to the journal guidelines

Thanks for your comment. We checked the accuracy and style of references

Round 2

Reviewer 2 Report

The authors replied to all comments. Therefore, according to my opinion, the paper can be accepted for pubblication in Nutrients Journal.

Author Response

Thank you